# The transcriptional landscape analysis of basal cell carcinomas reveals novel signalling pathways and actionable targets

Ivan V Litvinov ⓘ, Pingxing Xie, Scott Gunn, Denis Sasseville ⓘ, Philippe Lefrançois ⓘ

**Basal cell carcinoma (BCC) is the most common skin cancer and human malignancy. Although most BCCs are easily managed, some are aggressive locally, require Mohs micrographic surgery, or can even metastasize. In the latter, resistance to Sonic Hedgehog inhibitors may occur. Despite their frequent occurrence in clinical practice, their transcriptional landscape remains poorly understood. By analyzing BCC RNA sequencing data according to clinically important features (all BCCs versus normal skin, high-risk versus low-risk BCCs based solely on histopathological subtypes with aggressive features, advanced versus non-advanced BCCs, and vismodegib-resistant versus vismodegib-sensitive tumors), we have identified novel differentially regulated genes and new targetable pathways implicated in BCC tumorigenesis. Pathways as diverse as *IL-17*, *TLR*, *Akt/PI3K*, cadherins, integrins, *PDGF*, and *Wnt/β-catenin* are promising therapeutic avenues for local and systemic agents in managing this common malignancy, including through drug repurposing of existing medications. We experimentally validated several of these targets as biomarkers in our patient-derived cohort of primary BCC tumors.**

## Introduction

Basal cell carcinoma (BCC) is the most common skin malignancy and the most frequent of all human cancers (1). The lifetime risk for BCC is estimated to be 30% in individuals with fair (Fitzpatrick I-III) skin (2). The hallmark of BCC pathogenesis is the abnormal, constitutive activation of the sonic hedgehog (Shh) pathway (3). In the autosomal dominant Nevoid BCC syndrome, also known as the Gorlin–Goltz syndrome, loss-of-function mutations in *PTCH1* (*Patched*), and less often in *PTCH2*, *SUFU*, and *SMO* (*Smoothened*) result in *GLI*-mediated unopposed transcription of Shh target genes (4). In sporadic BCCs, the central role of *PTCH1* mutations in carcinogenesis has been confirmed by exome sequencing (5). In

addition to the Shh pathway, *TP53* is suggested to play a role in sporadic BCC (6, 7), likely through direct inhibition of *GLI* transcription factors (8).

Recently, small molecules inhibiting the Shh pathway, including vismodegib (9) and sonidegib (10), have been approved by the Food and Drug Administration (FDA) for locally advanced and metastatic BCC. However, both agents are poorly tolerated because of severe side effects, and in most patients, the clinical response is partial (11). About 30% of patients either do not respond to Shh inhibitors or develop resistance to treatment and relapse (11). Primary and secondary resistance arise via several mechanisms that activate sonic hedgehog signalling: (1) *SMO* mutations, either affecting the vismodegib binding pocket or allosterically, (2) mutations in genes downstream of *SMO*, such as *GLI2* or *SUFU*, and/or (3) via BCC cell identity switch towards a stem cell–like transcriptional program (12, 13, 14). There is currently a need to identify novel potentially targetable pathways outside of Shh signalling for treating advanced BCC.

Except for the Shh pathway, the transcriptional landscape of BCC remains incompletely understood. Microarray analyses identified up-regulation of *Wnt* signalling in addition to Shh (15). Other cell processes with higher expression levels in BCC than in normal skin include transcription, cell proliferation, cell metabolism, and apoptosis pathways (15). Recently, three studies have used exome sequencing analyses to identify driver mutations in BCC that were then validated by whole-genome RNA sequencing (RNA-Seq) analyses (7, 12, 13). In Bonilla et al, BCC was primarily driven by the Shh pathway, and additional driver mutations were found in several other genes resulting in *N-Myc* and *Hippo-YAP* pathway up-regulation (7). In Atwood et al and in Sharpe et al, molecular mechanisms of vismodegib treatment resistance were investigated, and RNA-Seq data confirmed transcriptional up-regulation of Shh downstream targets, including *GLI1* and cell proliferation marker *MKI67* (12, 13).

In this study, we explore the transcriptional landscape of BCC by pooling and re-analyzing publicly available RNA-Seq data to identify novel signalling pathways with clinical, pathophysiological, and therapeutic implications. We validated those findings experimentally in a patient-derived cohort of BCC tumors originating from dermatology clinics.

Division of Dermatology, Department of Medicine, McGill University, Montreal, Canada

Correspondence: philippe.lefrancois2@mail.mcgill.ca; ivan.litvinov@mcgill.ca

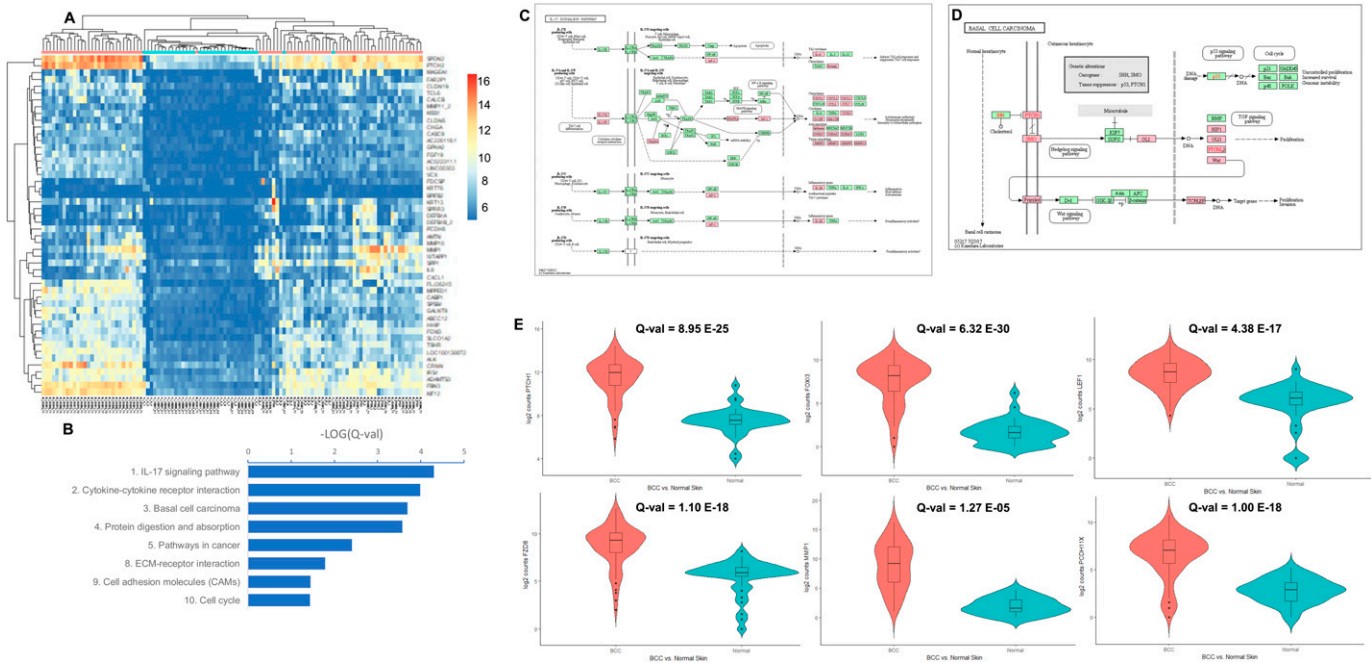

**Figure 1. All basal cell carcinoma (BCC) versus normal skin samples: up-regulated.**
**(A)** Unsupervised hierarchical clustering based on the top 50 most up-regulated genes in all BCCs (pink) versus normal skin controls (blue). A color key refers to gene expression in normalized $\log_2$ (pseudocounts). **(B)** Selected up-regulated KEGG pathways in BCC compared with normal skin. A false discovery rate Q-value cutoff <0.05 was used. The logarithm of negative (Q-value) is plotted. **(C, D)** Up-regulated genes in BCC overlaid on KEGG "IL-17 signalling pathway" (C) and "basal cell carcinoma" (D). In red are significant target genes. **(E)** Violin plots representing the distribution of gene expression, shown in $\log_2$ (pseudocounts), for all BCC (red) versus normal skin (turquoise). From left to right and top to bottom, plots for *PTCH1*, *FOXI3*, *LEF1*, *FZD8*, *MMP1*, and *PCDH11X* are displayed. Q-values are displayed.

# Results

### BCC versus normal skin

After re-analyzing RNA-Seq data of 75 BCC samples and 34 normal skin samples from the three publicly available datasets (Table S1 for breakdown), we have identified a total of 2,588 genes that were up-regulated and another 2,651 genes that were down-regulated in BCC compared with normal skin (Table S2).

To identify potential biomarkers for distinguishing BCC from normal skin, we performed hierarchical cluster analysis using the expression levels of the top 50 up-regulated genes. Three clusters were observed: the first cluster included only BCC samples (Fig 1A; 29/29), the second cluster included the remaining BCC samples plus two normal skin samples (Fig 1A; 41/43), and the third cluster consisted mostly of normal skin samples (Fig 1A; 32/37). Overall group membership was 93.5%, with a V-Measure of 0.48 (homogeneity = 0.66, completeness = 0.38).

After that, we focused the analyses on identifying significantly up-regulated or down-regulated pathways in BCC. Not surprisingly, Kyoto Encyclopedia of Genes and Genomes (KEGG) "BCC pathway" was found to be up-regulated ($P = 2.0 \times 10^{-04}$) in our analysis. KEGG "BCC pathway" is akin to the full Shh signalling pathway with a downstream Wnt pathway. Other up-regulated pathways included IL-17 signalling ($P = 4.9 \times 10^{-05}$), cytokine–cytokine receptor interactions ($P = 1.0 \times 10^{-04}$), protein digestion and absorption ($P = 2.7 \times 10^{-04}$), pathways in cancer ($P = 3.9 \times 10^{-03}$), extracellular matrix–

receptor interactions ($P = 0.016$), cell adhesion molecules ($P = 0.036$), and cell cycle ($P = 0.036$) (Fig 1B and Table S3). Up-regulated IL-17 pathway genes were mostly downstream (chemokines *CXCL1*, *CXCL2*, *CXCL5*, *CCL2*, *CCL5*, and Eotaxin; cytokines *COX2*, *IL-4*, *IL-1β*, and *G(M)-CSF*; defensins and other antimicrobials; matrix metalloproteinases *MMP1*, *MMP3*, *MMP9*, and *MMP13*) (Fig 1C), possibly representing tumoral inflammation in ulcerated or larger BCC. In comparison, up-regulated BCC pathway genes were distributed throughout the pathway (Fig 1D). Among KEGG pathways in cancer, up-regulated genes in Shh and Wnt signalling pathways were broadly distributed, highlighting again their role and interplay in BCC pathogenesis, along with genes inducing a block of differentiation (Fig S1). Up-regulated PANTHER pathways include cadherin signalling ($P = 6.6 \times 10^{-14}$), Wnt signalling ($P = 5.2 \times 10^{-08}$), and integrin signalling ($P = 0.036$) (Fig S2 and Table S4). RNA expression levels of several up-regulated genes in different signalling pathways are displayed using violin plots (Fig 1E).

Down-regulated KEGG pathways in BCC tumors included peroxisome proliferator–activated receptor (PPAR) signalling ($P = 3.0 \times 10^{-11}$), metabolic pathways ($P = 5.4 \times 10^{-09}$), fatty acid metabolism ($P = 6.3 \times 10^{-06}$), and retinol metabolism ($P = 1.5 \times 10^{-05}$) (Fig 2A and Table S5), indicating an overrepresentation of lipid metabolic pathways. Down-regulated genes were broadly distributed throughout the PPAR pathway (Fig 2B) and the retinol metabolic pathway (Fig 2C). In addition, strongly down-regulated Reactome pathways included keratinization ($P = 1.2 \times 10^{-43}$), metabolism of lipids and lipoproteins ($P = 2.2 \times 10^{-26}$) and formation of the cornified envelope ($1.1 \times 10^{-06}$) (Table S6).

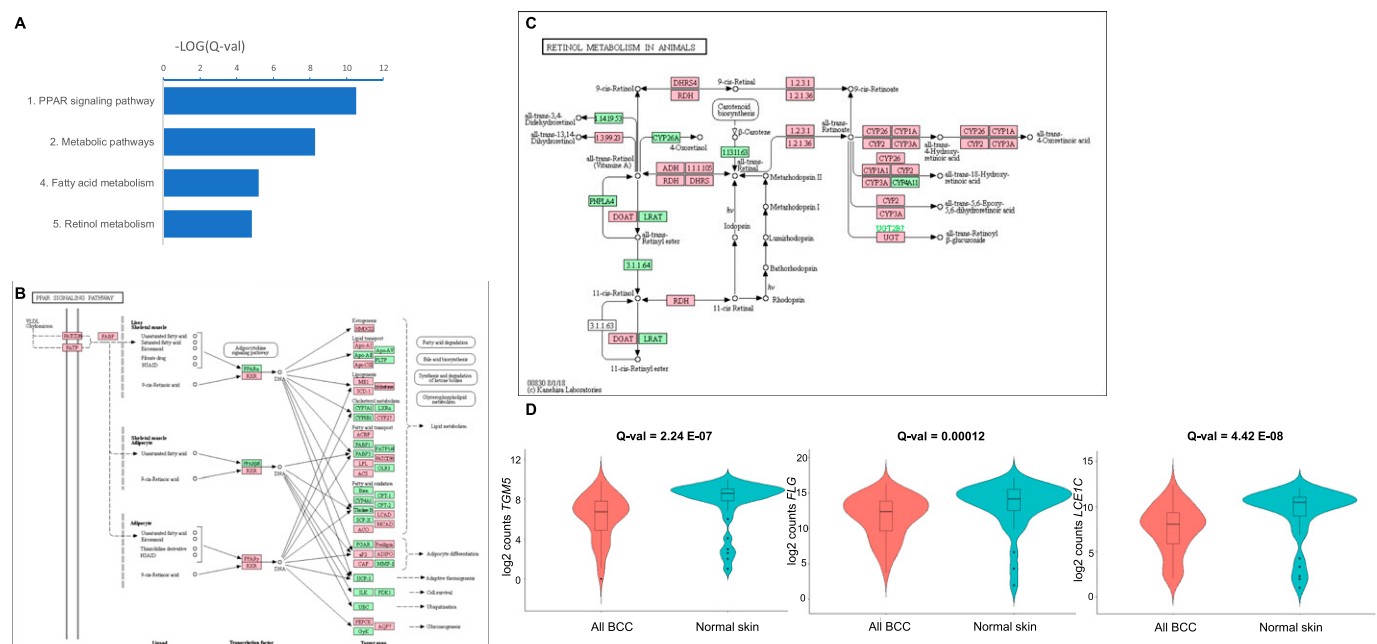

**Figure 2. All basal cell carcinomas (BCCs) versus normal skin samples: down-regulated.**
**(A)** Selected down-regulated KEGG pathways in BCC compared with normal skin. A false discovery rate Q-value cutoff <0.05 was used. The logarithm of negative (Q-value) is plotted. **(B, C)** Down-regulated genes in BCC overlaid on KEGG "peroxisome proliferator-activated receptor signalling pathway" (B) and "Retinol metabolism in animals" (C). In red are significant target genes. **(D)** Violin plots representing the distribution of gene expression, shown in log₂ (pseudocounts), for all BCC (red) versus normal skin (turquoise). From left to right, plots for *TGM5*, *FLG*, and *LCE1C* are displayed. Q-values are displayed.

Examples of RNA expression levels for down-regulated genes implicated in keratinization are displayed using violin plots (Fig 2D).

### Validation in McGill BCC sample cohort

We have obtained 15 simple, non-advanced BCC tumors and three normal skin samples originating from patients encountered in dermatology clinics at the McGill University Health Centre. cDNA generated from freshly obtained, liquid nitrogen snap-frozen tissues was analyzed by quantitative real-time reverse-transcription PCR (qRT-PCR) to determine the expression levels of 19 differentially expressed genes that spanned several enriched pathways in the previously detailed genomic analyses (Table S7). Overall, 13 of 15 BCC samples showed normalized enrichment above twofold for at least 50% of qRT-PCR–tested genes (heat map summarizing results in Fig 3); this simple approach yielded a sensitivity of 0.87 and a specificity of 1.0 for diagnosing a BCC (Fig S3A; AUC = 0.93). Receiver operator curves for individual genes as potential biomarkers are presented in Fig S3B. One gene was not validated: *CRNN*. We confirmed the down-regulation of *KRTAP11-1*, a gene involved in keratinization. As expected, key players in the sonic hedgehog pathway *SMO* and *PTCH1* were found to be up-regulated in all BCC samples compared with normal skin, so was *PTCH2* in most BCC. Few Wnt pathway genes were broadly up-regulated, such as *LEF1*, *FZD8*, and *DACT1*. IL-17 pathway downstream targets that were up-regulated include cytokines such *CXCL9* and matrix metalloproteinases involved in tumoral inflammation such as *MMP1* and *MMP10*. Cell cycle genes such as *CCND2* and *CDC6* also showed increased expression.

### High-risk BCCs (based solely on histopathological subtypes with aggressive features) versus low-risk BCCs

BCC tumors can present with a wide range of clinical and histopathological presentations, which may render them high risk for local recurrence and aggressive tissue destruction. Features such as body locations, patient characteristics (immunosuppression, genetic syndromes, radiotherapy, etc.), and tumor characteristics (histopathological subtypes with aggressive features, margin positivity, recurrent tumor, etc.) all have the potential to influence BCC risk level. The 2012 American Academy of Dermatology, American College of Mohs Surgery, American Society for Dermatologic Surgery Association, and the American Society for Mohs Surgery appropriate use criteria for Mohs micrographic surgery help identify clinical scenarios most likely to benefit from Mohs surgery, in the context of limited accessibility (16). BCC with high-risk features are usually managed by Mohs micrographic surgery, which was shown to significantly reduce the recurrence rate over more traditional medical and/or surgical approaches (17).

For subsequent analyses, in this study, we consider only the Mohs appropriate use criterion of "tumor characteristics – histopathological subtypes with aggressive features" (16). Thus, BCC tumors with superficial and nodular (especially on a low-risk body area) patterns would be considered low risk, when compared with high-risk BCC tumors with histological subtypes indicating a more

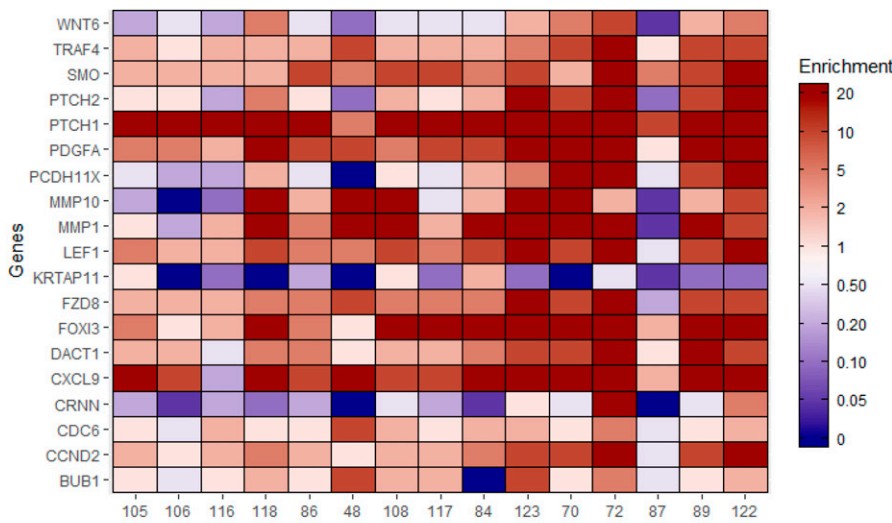

**Figure 3. Quantitative reverse-transcription PCR (qRT-PCR) for selected genes in McGill basal cell carcinoma (BCC) sample cohort.**
The heat map summarized the normalized enrichments for BCC samples from the McGill cohort. Normalized enrichment ratios (to housekeeping genes and to normal skin samples) are plotted. A normalized enrichment ratio of 1 indicates no enrichment over normal skin controls after normalization using housekeeping genes. A color key refers to the normalized enrichment ratio along predetermined ranges (<1 = blue; >1 = red). BCC patient samples are represented by their anonymous identifier on the x-axis and tested genes are on y-axis (Table S7).

aggressive biological behavior, such as morpheaform, infiltrating, metatypical, keratotic, and micronodular.

In our analyses, we compared the transcriptome of six high-risk BCC samples with 38 low-risk BCC samples, based uniquely on the histopathological subtypes with aggressive features criteria from the 2012 appropriate use criteria for Mohs micrographic surgery (see Methods). A total of 67 genes were found to be up-regulated (using a fold change cutoff of 1.50) and no genes were down-regulated in high-risk BCC samples (Table S8).

Hierarchical clustering on high-risk versus low-risk BCC samples based on expression levels of all 67 samples yielded three major patterns: one large cluster consisting only of low-risk BCCs, few early branching outgroups with high-risk BCCs, and a mixed cluster of high-risk and low-risk tumors (Fig 4A). The mixed cluster comprised samples with overall moderate up-regulation of *SPHK1*, *MTHFD1*, *BMS1P20*, *PRMT6*, *ANGEL1*, *OLFML2B*, and *C1QTNF6*, without any being highly up-regulated. Clustering metrics were lower (V-measure = 0.28 with homogeneity = 0.52 and completeness = 0.19), whereas accurate group membership was 93.2%.

No pathways were differentially up-regulated. Hence, we performed Gene Ontology (GO) analysis. GO molecular functions showed mild enrichment for sequence-specific DNA binding, double-stranded DNA binding, and DNA-binding transcription activator activity, RNA polymerase II–specific ($P = 0.049$ for all three GO terms) (Fig 4B and Table S9). Analysis of GO biological processes again revealed mild enrichment for genes involved in regulation of activin receptor signalling and regulation of DNA-binding transcription factor activity ($P = 0.030$ for both) (Fig 4C and Table S10). RNA expression levels for several up-regulated genes in high-risk BCC subtypes are displayed using violin plots (Fig 4D).

### Advanced versus non-advanced BCCs

Advanced BCCs include locally advanced and metastatic tumors. According to recent estimates, 0.8% and 0.04% of BCC tumors are locally advanced and metastatic, respectively (18). Although locally advanced BCCs do not have formal definite criteria, they represent tumors for which surgery and/or radiotherapy are usually not curative or not appropriate (e.g., resulting in mutilation) (9).

Here, we compared the transcriptome of 11 advanced BCCs with 44 non-advanced BCCs. All advanced tumors satisfied the inclusion criteria given in the prior paragraph. A total of 729 genes were found to be up-regulated and 1,624 genes were down-regulated in advanced BCCs (Table S11).

Hierarchical cluster analysis was performed using the expression levels for the top 50 up-regulated genes. Three major groups were observed: one large cluster consisting purely of non-advanced BCCs, one early outgroup joined with a small cluster of three advanced BCCs, and a mixed cluster without strong sub-cluster associations between advanced and non-advanced tumors (Fig 5A). The mixed cluster comprised samples with moderate to moderate-high up-regulation of *COL1A1*, *COL1A2*, *COL3A1*, *FN1*, and *LUM1*, without a strong *SFRP2* expression. Clustering metrics were lower (V-measure = 0.35 with homogeneity = 0.50 and completeness = 0.27), whereas accurate group membership was 87.3%.

Up-regulated KEGG pathways in advanced BCC included pathways that were also found to be up-regulated when comparing all BCC tumors with normal skin samples: protein digestion and absorption ($P = 1.3 \times 10^{-08}$), extracellular matrix–receptor interactions ($P = 1.4 \times 10^{-08}$), *IL-17* signalling ($P = 0.0039$), and cytokine–cytokine receptor interactions ($P = 0.011$) (Figs 5B and S4 and Table S12). In addition, *PI3K-Akt* signalling ($P = 0.0008$), proteoglycans in cancer ($P = 0.020$), and TLR signalling ($P = 0.028$) pathways were also up-regulated in advanced BCCs (Fig 5B and Table S12). Up-regulated *PI3K-Akt* pathway genes mostly comprised receptors and their ligands (Fig 5C). Up-regulated TLR receptor pathway genes also include receptors, their cognate ligands, and downstream cytokines (Fig 5D). Up-regulated PANTHER pathways included integrin signalling members ($P = 1.5 \times 10^{-07}$) (Fig S5 and Table S13). RNA expression levels for several up-regulated genes in different signalling pathways are displayed using violin plots (Fig 5E).

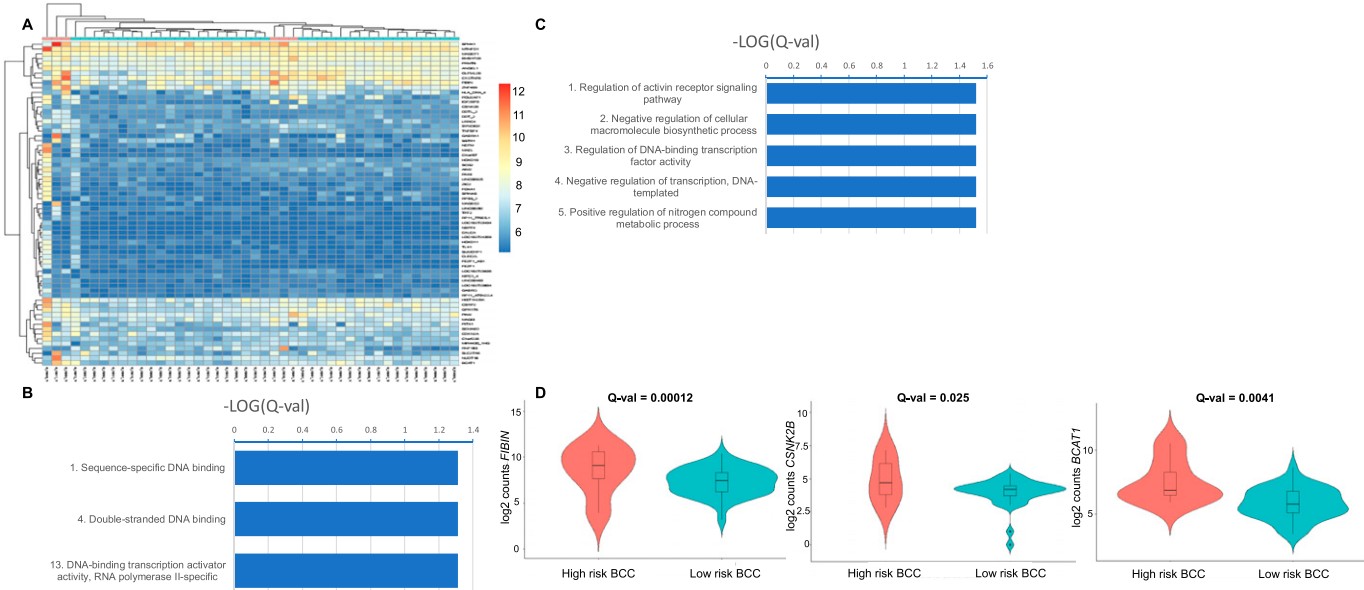

**Figure 4. High-risk basal cell carcinomas (BCCs) versus low-risk BCCs based on histopathological subtypes with aggressive features.**
**(A)** Unsupervised hierarchical clustering based on up-regulated genes in high-risk BCC subtypes (pink) versus low-risk BCC subtypes (blue). A color key refers to gene expression in normalized $\log_2$ (pseudocounts). **(B, C)** Selected enriched molecular function (B) and biological process (C) Gene Ontology terms from up-regulated genes in high-risk BCCs based solely on histopathological subtypes. A false discovery rate Q-value cutoff <0.05 was used. The logarithm of negative (Q-value) is plotted.
**(D)** Representative violin plots representing the distribution of gene expression, shown in $\log_2$ (pseudocounts), for high-risk BCC subtypes (red) versus low-risk BCC subtypes (turquoise). From left to right, plots for *FIBIN*, *CSNK2B*, and *BCAT1* are displayed. Q-values are displayed.

Pathway analysis for down-regulated genes in advanced BCC failed to reveal pathways with clear roles in advanced tumor pathophysiology. These included nicotinic acetylcholine receptor signalling pathway ($P = 0.036$) from the PANTHER database (Table S14), and muscle contraction ($P = 0.016$), cytochrome P450 ($P = 0.017$), xenobiotics ($P = 0.036$), and stimuli-sensing channels ($P = 0.036$) from the Reactome database (Table S15).

### Vismodegib-resistant versus vismodegib-sensitive BCCs

Shh inhibitors such as vismodegib and sonidegib are FDA-approved targeted therapies for advanced BCCs (9, 10). However, about 30% BCC tumors do not respond to Shh inhibitors (11).

The transcriptome of 20 vismodegib-resistant BCCs were compared with five vismodegib-sensitive BCCs. Clinical response to vismodegib as defined by the Response Evaluation Criteria in Solid Tumors guidelines was the end point for vismodegib sensitivity (partial response or complete response) versus resistance (progressive disease or stable disease) (12, 13). A total of 12 genes were up-regulated in vismodegib-resistant tumors and 67 genes were down-regulated (Table S16).

We performed hierarchical clustering on vismodegib-resistant versus vismodegib-sensitive BCCs based on the expression levels of all 79 differentially regulated genes (Fig 6A). Despite a limited number of differentially expressed genes, a group membership score of 100% was achieved when considering one large cluster only consisting of vismodegib-resistant tumors, and one looser association of an early outgroup of vismodegib-sensitive BCCs joined to a small cluster of vismodegib-sensitive BCCs. Clustering metrics were excellent, with a V-Measure of 0.84 (homogeneity = 1 and completeness = 0.72).

The only up-regulated pathway in vismodegib-resistant BCC was *Wnt/β-catenin* signalling ($P = 0.0080$; BioCarta), driven by the overexpression of *FSTL1* and *DACT1* (Fig 6B and Table S17; violin plots). RNA expression levels of several up-regulated genes in vismodegib-resistant BCCs are displayed using violin plots (Fig 6C).

Down-regulated Reactome pathways included muscle contraction ($P = 0.046$) and diseases associated with O-glycosylation of proteins ($P = 0.048$) (Table S18). Analysis of GO cellular functions showed mild enrichment for voltage-gated calcium channel complex and related ontology terms ($P = 0.042$) (Table S19).

To provide additional robustness regarding the up-regulation of *Wnt/β-catenin* signalling in vismodegib-resistant BCCs, we compared vismodegib-resistant tumors to untreated, vismodegib-naïve BCCs instead of vismodegib-treated, vismodegib-sensitive BCCs. Low-risk BCCs represent most of the vismodegib-naïve BCCs. Gene expression analyses and pathway enrichment analyses were performed as for other comparisons. Despite differences in biological and clinical behavior between vismodegib-naïve and vismodegib-resistant BCCs, the up-regulation of *Wnt/β-catenin* signalling in vismodegib-resistant BCCs remained ($P = 0.016$; PANTHER). Overexpression of *FSTL1* (fold change = 1.92; false discovery rate [FDR]-adjusted *P*-value = 0.0002) and *DACT1* (fold change = 1.91; FDR-adjusted *P*-value = 0.007) in vismodegib-resistant BCC persisted (Fig 6D; violin plots).

## Discussion

In this study, we further confirm the importance of increased Shh pathway activity for BCC tumorigenesis. We have summarized

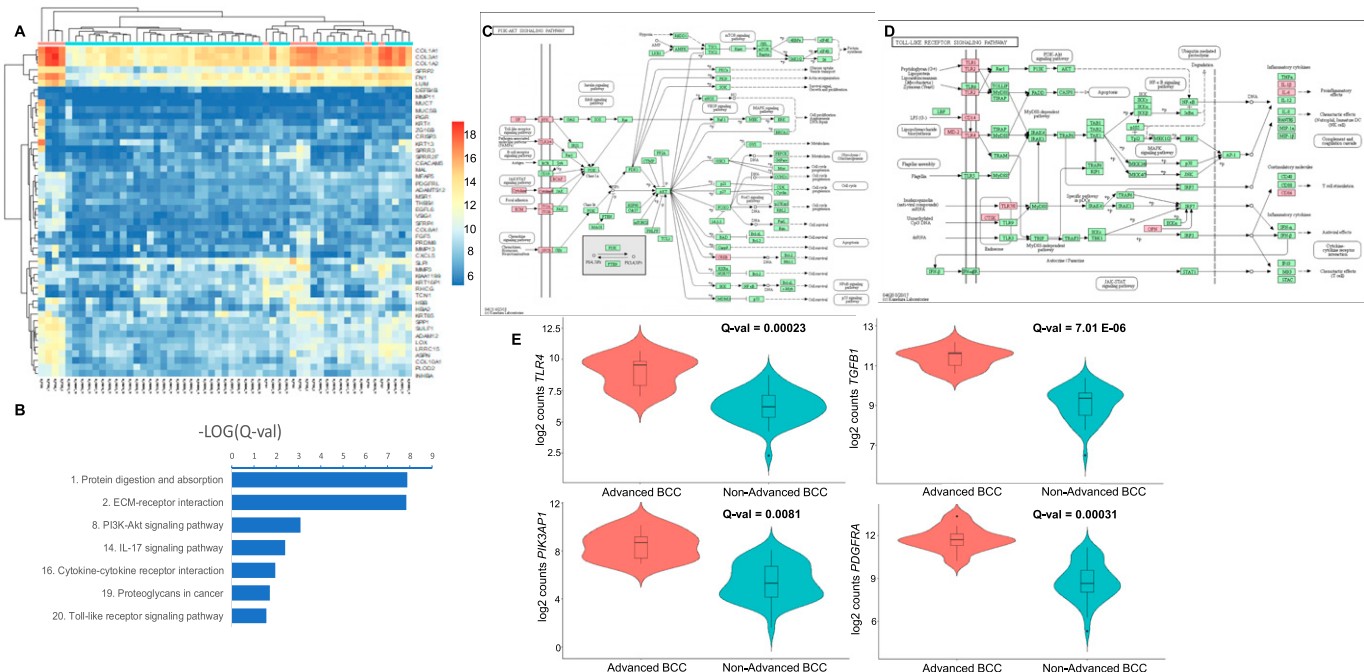

Figure 5. Advanced versus non-advanced basal cell carcinomas (BCCs).
(A) Unsupervised hierarchical clustering based on the top 50 most up-regulated genes in advanced BCCs (pink) versus non-advanced BCCs (blue). A color key refers to gene expression in normalized log$_2$ (pseudocounts). (B) Selected up-regulated KEGG pathways in advanced BCCs compared to non-advanced BCC. A false discovery rate Q-value cutoff <0.05 was used. The logarithm of negative (Q-value) is plotted. (C, D) Up-regulated genes in advanced BCCs overlaid on KEGG "PI3K/Akt signalling pathway" (C) and "TLR signalling pathway" (D). In red are significant target genes. (E) Violin plots representing the distribution of gene expression, shown in log$_2$ (pseudocounts), for advanced BCC (red) versus non-advanced BCC (turquoise). From left to right and top to bottom, plots for TLR4, TGFB1, PIK3AP1, and PDGFRA are displayed. Q-values are displayed.

selected up-regulated and down-regulated pathways according to clinical comparisons in Fig 7A. More importantly, we found other potentially actionable novel targets for BCC including *Wnt/β-catenin* and *IL-17* signalling pathways. A schematic diagram integrating many of these important signalling pathways for BCC is depicted in Fig 7B. Active canonical Wnt signalling acts through a feed-forward mechanism on the Shh pathway to sustain the latter's role in BCC (19). This contrasts to the antagonizing roles of *Wnt* and *Shh* signalling pathways during morphogenesis (20). In mice, combining a *Wnt* pathway inhibitor with vismodegib has led to a greater reduction in BCC tumor burden while promoting normal skin differentiation (14). Multiple *Wnt/β-catenin* pathway inhibitors are undergoing phase 0, I or II trials for BCC (21). It will be interesting to test their synergistic effect with *Shh* inhibitors in advanced BCCs in the future. *IL-17* signalling has been implicated in tumorigenic inflammatory responses (22) and has been associated with breast cancer progression (23). Currently, monoclonal antibodies targeting IL-17 or IL-17R proteins are indicated for moderate-to-severe psoriasis (24). IL-17 and IL-22 cytokines have been directly implicated in BCC and SCC progression and tumor growth in mouse xenografts, as well as progression, cell migration, and local invasion in BCC cancer cell lines (25). Another report confirmed up-regulation of both cytokines in peritumoral skin of BCC, correlating with the severity of the inflammatory infiltrate (26). Both studies have suggested IL-17 as a potential therapeutic target. From our analysis, for advanced BCC, there might be a potential benefit using

anti-IL-17 therapies for treating these tumors, with a favorable safety profile. This effect remains to be proven in clinical trials. For non-advanced BCC, a topical or an intratumoral formulation may show efficacy. Among other up-regulated pathways, cadherins, and integrins have roles in carcinogenesis. Loss of *E-Cadherin* expression and up-regulation of *N-Cadherin* increase epithelial-to-mesenchymal transition and thus metastasis through canonical *Wnt* signalling (27); the small molecule inhibitor ADH-1 has led to a disease control in advanced solid tumors during preclinical and phase I trials (27). STX-100, an anti-β6 integrin agent, is undergoing clinical trials for fibrotic diseases and cancers (28).

Among BCC tumors, lipid-related metabolic pathways that are essential to generate, maintain, and regenerate a fully functional keratinized epidermis appear to be down-regulated. For example, impaired lipid metabolism and down-regulation of PPAR pathways are associated with impaired wound healing (29). These alterations might contribute to the non-healing phenotype of certain BCCs.

A major limitation of this study is the lack of power because of small sample size. This affects more strikingly uncommon groups of tumors, such as vismodegib-treated, vismodegib-sensitive tumors (n = 5) and BCC with a high-risk histopathological type (n = 6). Other groups had at least 10 samples. In the hope of increasing robustness, for the comparison on vismodegib resistance, we also validated the major findings using the larger group of vismodegib-naïve BCCs as a control group, instead of vismodegib-treated, vismodegib-sensitive BCCs. Larger cohorts of patient samples might

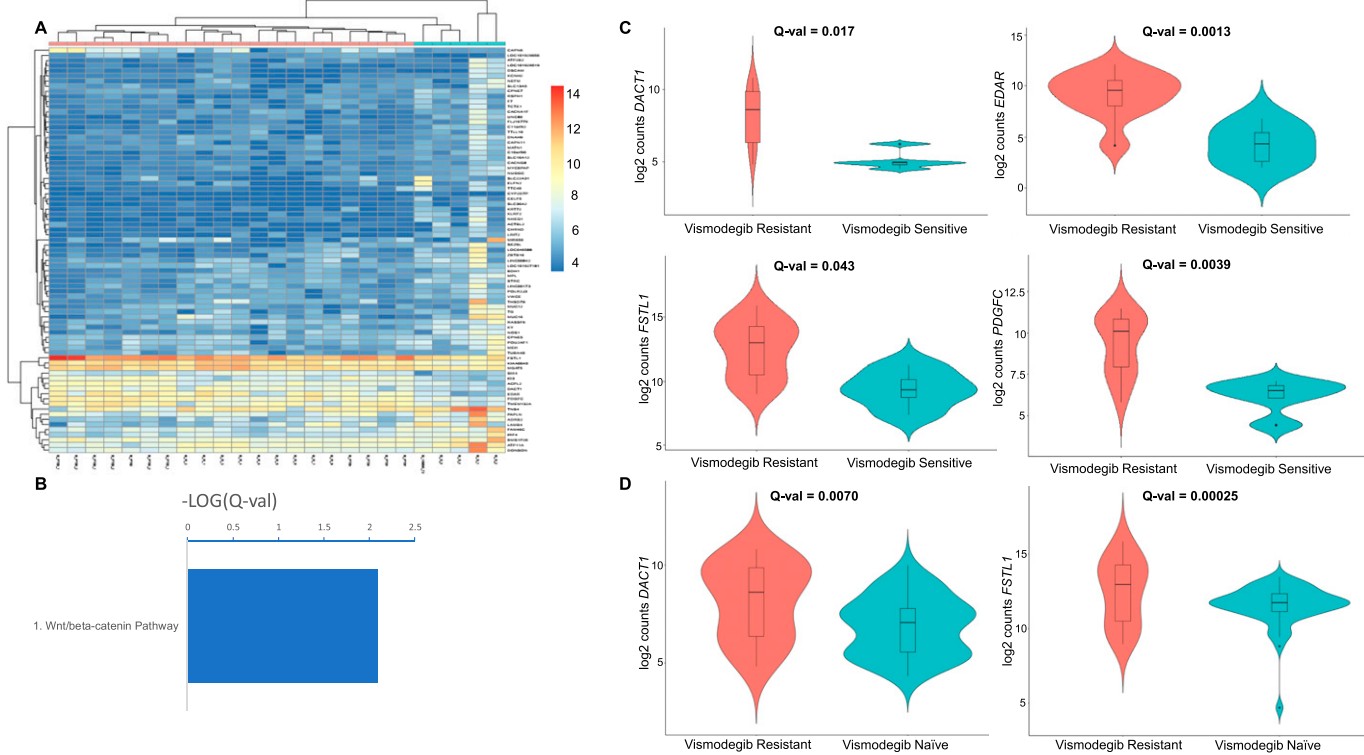

**Figure 6.   Vismodegib-resistant versus vismodegib-sensitive basal cell carcinomas (BCCs).**
**(A)** Unsupervised hierarchical clustering based on all differentially regulated (up and down) genes in vismodegib-resistant BCCs (pink) versus vismodegib-sensitive BCCs (blue). A color key refers to gene expression in normalized log$_2$ (pseudocounts). **(B)** Up-regulated BioCarta pathways in vismodegib-resistant BCCs compared with vismodegib-sensitive BCCs. A false discovery rate Q-value cutoff <0.05 was used. The logarithm of negative (Q-value) is plotted. **(C)** Violin plots representing the distribution of gene expression, shown in log$_2$ (pseudocounts), for vismodegib-resistant (red) versus vismodegib-sensitive BCCs (turquoise). From left to right and top to bottom, plots for *DACT1*, *EDAR*, *FSTL1*, and *PDGFC* are displayed. Q-values are displayed. **(D)** Violin plots representing the distribution of gene expression, shown in log$_2$ (pseudocounts), for vismodegib-resistant (red) versus vismodegib-naïve (untreated) BCCs (turquoise). From left to right, plots for *DACT1* and *FSTL1* are displayed. Q-values are displayed. Please note that in (D), vismodegib-naïve tumors are displayed as a comparative group.

shed light on additional biologically meaningful pathways that could be missed in the current comparisons.

Another limitation relates to missing key elements from the clinical information. For example, for advanced BCCs, it was not possible to know whether a selection bias might have been a reason why surgery and/or radiation were not performed, or whether patient and tumor factors such as the underlying location, immunosuppression status, and histopathological subtypes played a role in clinical decision-making.

Less than 100 genes were differentially regulated when comparing BCCs based on histopathological subtypes (67 up-regulated genes and 0 down-regulated gene) or on sensitivity to vismodegib (12 up-regulated genes and 67 down-regulated genes). We recognize that these comparisons suffer from a lack of power due to small sample sizes, as both feature a group with <10 tumors (six for high-risk histopathological subtypes, five for vismodegib-sensitive, vismodegib-treated tumors). Despite the limited number of genes, classification accuracy based on their expression levels was high, indicating that these genes could be used as potential biomarkers to distinguish clinical BCC tumors based on histopathological subtype (high-risk versus low-risk) or vismodegib resistance status (resistant versus sensitive). For high-risk BCCs based on histopathological subtypes with aggressive features, up-regulated genes

of interest included *CDKN2A*, whose germline mutations are implicated in hereditary malignant melanoma ([30]), *BCAT1*, an amino acid transaminase implicated in breast cancer progression through increased mTOR-mediated mitochondrial activity ([31]), and *AIM2*, a contributor to tumoral inflammation in cutaneous squamous cell carcinoma ([32]). Further studies on a large cohort of high-risk BCC tumors based on histopathological subtypes, for which samples are harder to obtain in a dermatology practice, may yield additional clues to important molecular pathways.

It was shown that residual resistant BCC tumor cells after vismodegib treatment in a mouse model can adopt a stem cell–like transcriptional profile, along with increased *Wnt* signalling ([14]). In vismodegib-resistant BCC samples, we observed an up-regulation of two *Wnt/β-catenin* pathway modulators, *FSTL1* and *DACT1*. *FSTL1* is an inflammatory protein. Blocking *FSTL1* via a monoclonal antibody suppresses metastasis and disease progression in several mouse tumor models ([33]). *DACT1* on the other hand stabilizes *β-catenin*, leading to disease progression in colon cancer ([34]). The up-regulation of *Wnt/β-catenin* pathway and of two of its modulators, *FSTL1* and *DACT1*, in resistant tumors is likely an intrinsic characteristic of vismodegib resistance, as it remains elevated compared with untreated, vismodegib-naïve BCCs. Among other up-regulated genes in vismodegib-resistant tumors, two have

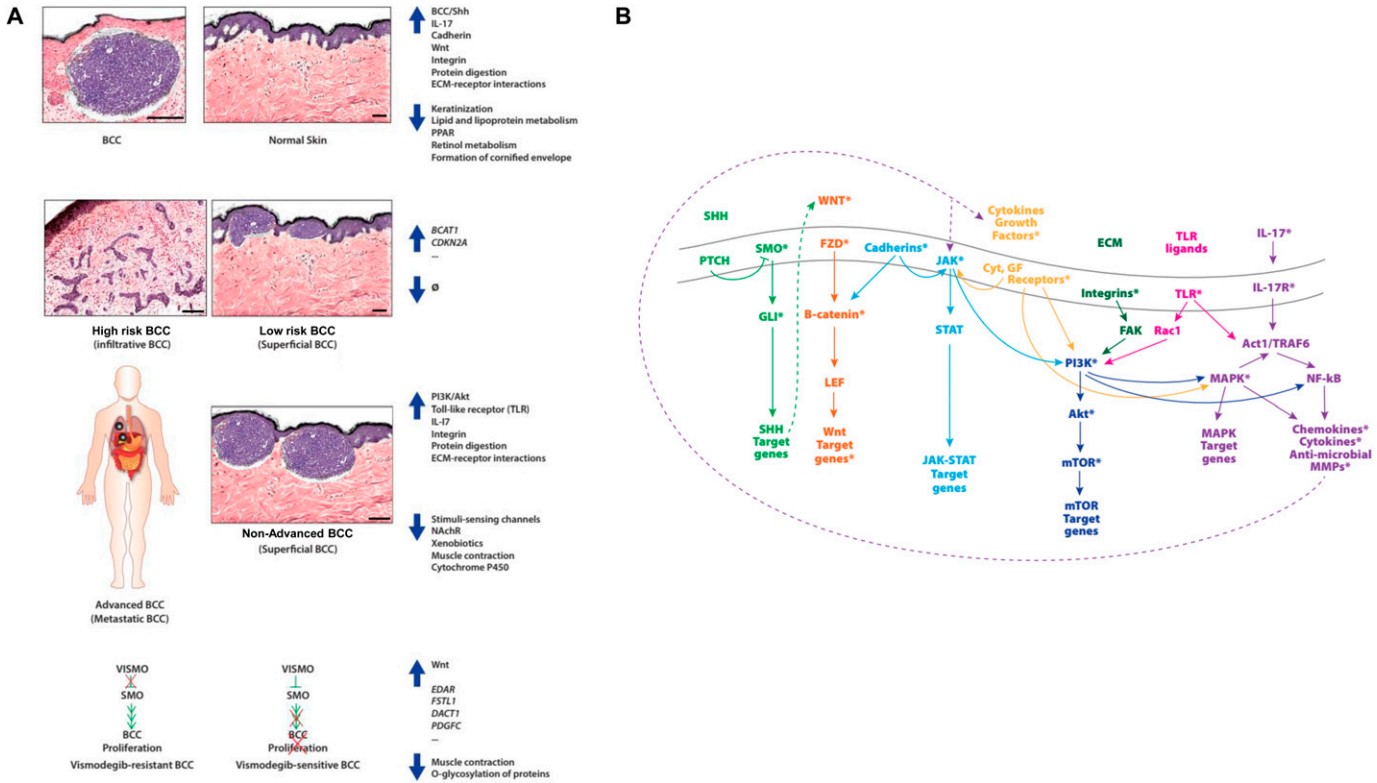

**Figure 7. Summary of differentially regulated and potentially targetable pathways in basal cell carcinoma (BCC).**
**(A)** Summary of enriched differentially regulated pathways and/or genes for all BCCs versus normal skin (top panel), high-risk versus low-risk BCCs (second top panel), advanced versus non-advanced BCCs (second bottom panel), and vismodegib-resistant versus vismodegib-sensitive BCCs (bottom panel). Scale bars on schematized histology sections represent approximately 200 μm. **(B)** Potential therapeutic targets for BCCs in select signalling pathways. Asterisks indicate specific molecules for which agents are either available or under development.

known functions, *EDAR* and *PDGFC*. *EDAR* mutations cause hypohidrotic ectodermal dysplasia (35). *PDGFC* encodes a platelet-derived growth factor that has an anti-apoptotic role in tumor-associated macrophages, thus contributing to tumorigenesis and malignant inflammation (36). Signalling through *PDGFC* can be abrogated by *PDGFR* inhibitors such as imatinib, sunitib, and pazopanib (37). Based on the results of our study, utility of *PDGFR* inhibitors should be investigated for vismodegib-resistant BCC tumors. Other pathways involved in vismodegib resistance may be uncovered after RNA-Seq data from a more substantial number of vismodegib-sensitive BCCs becomes available.

Advanced BCCs up-regulate extracellular matrix proteins and protein degradation machinery, which is consistent with their locally destructive behavior, their stromal reaction protecting them from the host's immune response, and their metastatic potential. Matrix metalloproteinases (*MMP*) are involved in tumoral inflammation, cancer cell proliferation, angiogenesis, and angioinvasion (38). *MMP* inhibitors with broad activity have not had the same success in human studies, when compared with mice (38); targeted *MMP* inhibitors are currently under investigation (38). In advanced BCC samples, *TLR* signalling and *PI3K/Akt* signalling were also found to be up-regulated. TLR agonists such as imiquimod are used to treat low-risk superficial BCC (39). However, in solid tumors, paradoxically, increased TLR signalling is linked to

immune suppression, tumor proliferation, survival, and metastasis (40). Except for *PIK3AP1/BCAP*, up-regulated *PI3K/Akt* pathway genes mostly encode receptors and their ligands, namely upstream of *PI3K* activation. *PIK3AP1* encodes the B-cell adapter for *PI3K* linking *CD19* and B-cell receptor in the *PI3K* pathway (41). It also activates the *NF-κB* pathway and is critical for *PI3K/Akt* activation through TLR signalling (42).

Genes identified from this BCC transcriptome analysis can be tested in routine biopsy samples obtained from patients in the clinics. They may be used as biomarkers for diagnostic or therapeutic stratification quite successfully. We have tested in silico findings from RNA-Seq data in a different cohort of patients presenting with BCCs. We confirmed, in our clinical samples, the value of key members of the *Wnt/β-catenin* and *IL-17* signalling pathways, cell cycle genes, as well as of important players of tumoral inflammation such as matrix metalloproteinases and cytokines, as molecular biomarkers. Overall, our success rates in the McGill validation cohort were ~95% for tested genes (18/19) and ~87% for BCC tumors (13/15). A larger validation cohort is likely needed to further characterize these biomarkers. Targeted therapies in BCCs are mostly administered systemically and reserved for advanced cases because of significant cost and adverse effects. Because <1% of BCC tumors are advanced (18), local delivery of targeted therapies (topical or intralesional) represents a promising avenue for

non-advanced BCCs. One such example is the development of topical sonic hedgehog inhibitors such as patidegib (43). Our study has provided evidence for other potential drug targets for both future local and systemic therapies.

# Materials and Methods

### Data acquisition

We obtained whole-genome RNA-sequencing (RNA-Seq) data of 75 BCC samples and 34 normal skin samples: 13 BCC samples and eight normal skin samples were from Atwood et al (Gene Expression Omnibus accession number GSE58377) (13), 51 BCC samples and 26 normal skin samples were from Bonilla et al (European Genome-phenome Archive EGA accession number EGAS00001001540) (7), and 11 BCC samples from Sharpe et al (EGA accession number EGAS00001000845) (12). A breakdown of samples and clinical data are presented in Table S1.

### Primary RNA-Seq analysis and differential gene expression analysis

Primary RNA-Seq analysis was performed as previously reported (44). Raw fastq files were curated to remove low-quality reads using FastX-Toolkit. Reads were aligned against the hg38 reference human genome using HISAT2 (45). Read counts were obtained using htseq-count (46), whereas transcripts per million reads were computed by Cufflinks (47). Median Transcript Integrity Number (medTIN) (48), an in silico measurement of RNA integrity that is similar to the experimentally derived RNA integrity number, have been generated for all samples (median medTIN = 77.12). There were no statistically significant differences between medTINs from different comparisons ($P > 0.05$; Mann–Whitney U test; Fig S6), including normal skin versus BCC (Fig S6A), low-risk versus high-risk BCC based solely on histopathological subtypes with aggressive features (Fig S6B), non-advanced versus advanced BCC (Fig S6C), and vismodegib-sensitive versus vismodegib-resistant BCC (Fig S6D). Batch effect was assessed and accounted for as described (49).

Tumor purity was determined by ESTIMATE (50) using gene expression data. We have divided BCCs into aggressive BCCs (advanced tumors + high-risk BCCs based on histopathological subtypes) and non-aggressive BCCs. Median tumor purities were 81.5% for aggressive BCCs versus 91.0% for non-aggressive counterparts, a statistically significance difference ($P < 0.001$; Whitney–Mann U test; Fig S7). Considering the biology of BCC, this is expected for several reasons. First, small tumors usually can be easily demarcated, both clinically and with dermoscopy. Second, advanced tumors are typically more inflammatory or may be ulcerated. Third, BCCs rely heavily on stromal elements, and are in fact characterized by a fibromyxoid stroma, especially for high-risk BCCs and BCCs undergoing progression (51). Both an inflammatory response and a stromal response are characteristic (52). Most gene expression-based algorithms for tumor purities consider tumor purity as pure tumoral component minus stromal abundance and immune infiltration (50).

To better visualize effects from tumor purity, we have performed Principal Component Analysis on gene expression and tumor purity. We have generated principal component analysis plots according to the first two principal components, with points sized according to tumor purity (higher tumor purity = larger size), for all four comparisons: normal skin versus BCC (Fig S8A), low-risk versus high-risk BCC based solely on histopathological subtypes with aggressive features (Fig S8B), non-advanced versus advanced BCC (Fig S8C), and vismodegib-sensitive versus vismodegib-resistant BCC (Fig S8D). We did not observe significant clustering of samples based primarily on tumor purity.

Differentially expressed genes were identified by edgeR (53) with fold change >2.0 (1.5 for the histopathological subtypes and vismodegib-naïve analyses) and FDR corrected for multiple hypothesis testing using Benjamini–Hochberg method <0.05. The following comparisons were designed: (1) BCCs versus normal skin (7, 12, 13), (2) high-risk BCCs with histological subtypes having aggressive features (morpheaform, micronodular, metatypical, keratotic, and infiltrative) versus low-risk BCCs (nodular, superficial, others) (7), (3) locally advanced and metastatic BCCs (advanced) versus non-advanced BCCs (7, 12), and (4) vismodegib-resistant versus vismodegib-sensitive BCCs (7, 12, 13). For comparison (2) (histopathological subtypes), the distinction is purely based on subtypes themselves, and not on patient outcomes nor on treatment selection, as these clinical data are not available. For visualization of individual gene expression, log 2 pseudo-counts for selected genes were plotted using R package ggplot2 (54).

### Downstream RNA-Seq analysis from gene lists

Enriched pathways and GO terms were determined using ToppFun (55), with FDR corrected for multiple hypothesis testing <0.05 (56). We considered the following reference databases for pathways: KEGG (57), PANTHER (58), Reactome (59), and National Cancer Institute's BioCarta. Hierarchical clustering was performed using either all differentially expressed genes, when the total number of genes was <100 or using the following procedure: among the top 100 most up-regulated genes as ranked by fold change, we selected the top 50 genes as ranked by logCPM to high raw RNA expression level. Heat maps according to the gene expression levels and samples were generated using R package heat map (60). V-measure, a clustering validity metric combining homogeneity of clusters and completeness of clusters, were determined for all heat maps (61).

### Patients and tissues

All patients were enrolled in this study with written informed consent and in accordance with the Declaration of Helsinki from McGill University Health Centre and affiliated hospitals (REB study #2018-4134 and #2018-3962). Fifteen BCC samples and three normal skin samples (fibroepithelial polyps, seborrheic keratosis) were freshly obtained, snap-frozen for gene expression analysis.

### Real-time quantitative reverse-transcription PCR (qRT-PCR)

RNA was extracted from tissues using Trizol (Thermo Fisher Scientific) and converted to cDNA using iScript cDNA synthesis kit (Bio-Rad). qRT-PCR was performed as previously described (62).

Normalized enrichments were calculated according to the Pfaffl method (63) as previously described (64), normalizing to normal skin control samples and to two housekeeping genes (*SDHA* and *B2M*). Primers are available upon request. Receiver operator curves were generated using R package pROC (65).

## Data Availability

Accession numbers of published RNA-Seq datasets are reported in the Materials and Methods section. Primers for qRT-PCR are available upon request.

## Supplementary Information

## Acknowledgements

We are very grateful to Dr. Frederic de Sauvage (Genentech) and Dr. Sergey Nikolaev (University of Geneva) for facilitating data sharing. We thank Ms. Ildiko Horvath for medical illustration. This work was supported by the Canadian Dermatology Foundation research grants to Dr. D Sasseville and Dr. IV Litvinov, and by the Fonds de la recherche du Québec – Santé to Dr. D Sasseville (#22648) and to Dr. IV Litvinov (#34753 and #36769). This research was enabled in part by support provided by Calcul Québec (www.calculquebec.ca), Compute Ontario (www.computeontario.ca), WestGrid (www.westgrid.ca), and Compute Canada (www.computecanada.ca).

### Author Contributions

IV Litvinov: resources, supervision, funding acquisition, validation, investigation, methodology, project administration, and writing—original draft, review, and editing.
P Xie: resources, formal analysis, investigation, methodology, and writing—original draft, review, and editing.
S Gunn: resources, formal analysis, validation, investigation, methodology, and writing—original draft, review, and editing.
D Sasseville: supervision, funding acquisition, investigation, project administration, and writing—original draft, review, and editing.
P Lefrançois: conceptualization, resources, data curation, formal analysis, supervision, funding acquisition, validation, investigation, visualization, methodology, and writing—original draft, review, and editing.

### Conflict of Interest Statement

The authors declare that they have no conflict of interest.

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
