## [Reviewer comments · Life Science Alliance]

Life Science Alliance

Transcriptional landscape of basal cell carcinomas: novel signaling pathways and actionable targets

Ivan Litvinov, Pingxing Xie, Scott Gunn, Denis Sasseville, and Philippe Lefrançois

DOI: <https://doi.org/10.26508/lsa.202000651>

Corresponding author(s): Philippe Lefrançois, McGill University Faculty of Medicine and Ivan Litvinov, McGill University Faculty of Medicine

Review Timeline:

Submission Date:	2020-01-20
Editorial Decision:	2020-02-27
Revision Received:	2021-02-27
Editorial Decision:	2021-04-09
Revision Received:	2021-04-23
Accepted:	2021-04-26

Scientific Editor: Shachi Bhatt

Transaction Report:

February 27, 2020

Re: Life Science Alliance manuscript #LSA-2020-00651-T

Dr. Ivan V Litvinov
McGill University Faculty of Medicine
Department of Medicine, Division of Dermatology
E02.6236 Royal Victoria Hospital
1001 Decarie Blvd
Montreal, Quebec H4A 3J1
CANADA

Dear Dr. Litvinov,

Thank you for submitting your manuscript entitled "Transcriptional landscape of basal cell carcinomas: novel signaling pathways and actionable targets" to Life Science Alliance. The manuscript was assessed by expert reviewers, whose comments are appended to this letter.

As you will see, the reviewers are supportive of publication of your work, pending revision to ensure robustness, that your conclusions are supported by the data provided, and to leave room for alternative explanations. We would thus like to invite you to submit a revised version of your manuscript to us, addressing the individual points raised by reviewer #1 and #2.

Thank you for this interesting contribution to Life Science Alliance. We are looking forward to receiving your revised manuscript.

Sincerely,

B. MANUSCRIPT ORGANIZATION AND FORMATTING:

Reviewer #1 (Comments to the Authors (Required)):

The manuscript of Dr P. Lefrançois et al., entitled: "The transcriptional landscape analysis of basal cell carcinomas reveals novel signaling pathways and actionable targets" presents analysis of

publically available RNAseq data from 75 BCC and 34 samples of normal skin. Analyses of Lefrancois et al., tackled different aspects of BCC heterogeneity and reveal differentially expressed genes and pathways in BCC vs. skin and in BCC associated with aggressiveness.

Comments:

One of the limitations of this analysis is associated with the very small number of samples representing rare subtypes. This has to be discussed in the text.

How did the authors assess the quality of the sequenced RNA? What was the Transcript Integrity Number (TIN) for each RNA sample, and was it taken into account?

Samples from Atwood and Sharpe may be overlapping. Was uniqueness of the sample origin verified?

In general, most aggressive tumors are frequently bigger and with better tumor purity (TP) than benign lesions, this may explain "enrichment" of tumor markers associated with aggressiveness. A graph representing distribution of tumor purity (TP) would be useful. It is also possible to compare TP estimates based on RNA with TP from mutational data. TP is an important confounding factor which can impact sample clusterization. PCA may help to reveal and account for the TP component.

In figure 4c, high-risk BCC do not seem to form a separate cluster, low-risk BCC also represent one pure group, and other - includes both low and high risk BCC. There is no clear message about the differences at RNA level between these subgroups. How classification accuracy was calculated? Same comment is relevant for figure 5.

One of the interesting findings is the upregulation of WNT pathway in vismo-resistant BCC. Would this result remain if vismo-resistant BCC contrasted with naïve to treatment BCC, majority of which are expected to respond to vismodegib? This would help to improve statistics.

Labeling of the figures does not correspond to references in the text.

Reviewer #2 (Comments to the Authors (Required)):

This is a very interesting manuscript from an established group of investigators. The authors identified IL-17 pathways previously not shown in various clinical types of BCC. While there is a certain novelty of their research, there are several minor points that need to be addressed.

p. 5, the second paragraph, about lipid-related metabolic pathways and "this can be reflected in the tendency for BCC to ulcerate and/or easily bleed." This statement is purely hypothetical and should be substantiated by the appropriate references. I would recommend moving it to a discussion from the results.

All violin plots are lacking statistical analysis of the differences. Please provide the appropriate statistics on the charts.

p.6 "Validation in McGill BCC samples cohort." The authors identified 19 differentially-expressed genes that they try to validate. Traditionally, the validation involves ROC curves and AUC comparison between the previous set and new set of samples. It is tough to judge if those results are valid by a cluster analysis.

p.7 "High risk BCC with histopathological subtype requiring Mohs..."

"BCC tumors with superficial and nodular histopathological patterns are usually considered low-risk." This statement is incorrect. The nodular BCC are never considered to be a low-risk, and for that reason, none of the topical treatments are FDA approved for the treatment of nodular BCC. There are very strict criteria for Mohs selection of NMSC, which include not only histological type (by the way, basosquamous BCC is not one of them, but metatypical and keratotic is included), but also area of the body, the patient characteristics, and the positive margin on the recent excision/biopsy (Connolly et al AAD/ACMS/ASDSA/ASMS 2012 appropriate use criteria... JAAD, 2012). Thus, it is not entirely clear to me what criteria were used for high-risk BCC samples. Samples that were obtained from Mohs? Some histological features of BCC? In any case, the samples may be too heterogenous, and I am questioning the utility of such separation.

p.8 "Advances vs. non-advanced BCC" Please clarify the clinical and histological characteristics of your advanced BCC. What was the reason not to perform the surgery or radiation? Size of the tumors, age of the patients, location? All those features may contribute to the selection bias of your BCC in the advanced group.

p.9 how vismodegib-sensitive and resistant tumors were established? Ex vivo? Clinical response? please clarify

p. 10 "for advanced BCC, anti-IL17 therapy may have activity..." the reference that was provided at the end of that sentence does not support this statement. Please, supply the correct reference.

p.11 discussion about natalizumab. I am not sure why there is a need to discuss JC virus reactivation in the manuscript about BCC.

Figure 7A. The nodular BCC was shown on a histological picture, and the title underneath stated "conservative management (superficial BCC)". First of all, it is not a superficial BCC; second of all, the conservative management (whatever that means) is not appropriate for nodular BCC. The same goes for the next picture. The same BCC, the title underneath stated "simple BCC (superficial BCC)": again not superficial BCC on the picture but nodular and not sure what "simple" means. Please make some corrections.

Reviewer #3 (Comments to the Authors (Required)):

Lefrancois et al. perform an in-depth analysis of gene expression of basal cell carcinomas (BCC) to reveal novel pathways that may serve as targets for future treatment. The investigators studied a number of different BCC variant transcriptomes that include low-risk, high-risk, advanced, vismodegib sensitive and vismodegib resistant tumors. The team analyzed differences in gene expression utilizing RNAseq from publically available dataset. The combination of datasets yield robust findings for the different variants of BCC that highlight pathways important in the pathogenesis of BCC. The authors analyzed gene sets for relevant pathways using KEGG, PANTHER, Reactome and BioCarta. The genes identified revealed pathways that included IL-17,

TLR, Akt/PI3K, cadherins, integrins, and the Wnt/beta-catenin pathways. A strength of the study is that the investigators were able to validate a subset of genes by qRT-PCR from primary BCC samples. The analysis is novel and innovative, and new targets for treatment of BCC may potentially be developed.

Reviewers' comments:

Reviewer #1: We are thankful for the astute comments by Reviewer #1 that led us to resubmit an improved manuscript.

The transcriptional landscape analysis of basal cell carcinomas reveals novel signaling pathways and actionable targets" presents analysis of publically available RNAseq data from 75 BCC and 34 samples of normal skin. Analyses of Lefrancois et al., tackled different aspects of BCC heterogeneity and reveal differentially expressed genes and pathways in BCC vs. skin and in BCC associated with aggressiveness.

R1-C1

One of the limitations of this analysis is associated with the very small number of samples representing rare subtypes. This has to be discussed in the text.

Our response: We agree that this is a major limitation for analyses representing rare subtypes. We have added an additional paragraph describing this limitation.

"A major limitation of this study is the lack of power due to small sample size. This affects more strikingly uncommon groups of tumors, such as vismodegib-treated, vismodegib-sensitive tumors (n = 5) and BCC with a high-risk histopathological type (n = 6). Other groups had at least ten samples. In the hope of increasing robustness, for the comparison on vismodegib resistance, we also validated the major findings using the larger group of vismodegib-naïve BCCs as a control group, instead of vismodegib-treated, vismodegib-sensitive BCCs. Larger cohorts of patient samples might shed light on additional biologically meaningful pathways that could be missed in the current comparisons." Lines 276-283

R1-C2

How did the authors assess the quality of the sequenced RNA? What was the Transcript Integrity Number (TIN) for each RNA sample, and was it taken into account?

Our response: We have used standard RNA-Seq processing, including removal of low quality reads using FastX Toolkit as stated in the methods. The most commonly used method for RNA quality evaluation of RNA-Seq experiments is the experimental determination of RNA integrity numbers (RIN), but we do not have access to these data due to the design of our study. We had not initially computed Transcript Integrity Number (TIN) [Wang et al., 2016], since these were not reported in the original articles.

We have computed TINs for all samples. The median median TIN (medTIN) was 77.12 and the mean medTIN was 68.62. We have plotted medTINs on a per-sample basis, according to the 4 comparisons performed in the text, and assessed whether distributions differed between compared conditions using Mann-Whitney U tests (e.g. BCC vs. normal, advanced vs. non-advanced, ...). No statistically significant differences were found. We have added a paragraph in the methods and a Supplementary Figure 6.

We did not account for TINs within our analyses, for 2 reasons. 1- We already accounted for batch effect and tumor purity in our analyses (see below). 2 - There are no consensus methods to account for TINs with wide acceptance/use. As such, it is not used by large consortia such as ENCODE. Some algorithms have started to emerge to correct for TINs (DegNorm, flexiMAP and others), but have not been widely adopted yet. We hope that there is a more streamlined approach in the future to account for TINs.

- Supplementary Figure 6

"Median Transcript Integrity Number (medTIN) [Wang et al., 2016], an *in silico* measurement of RNA integrity that is similar to the experimentally-derived RNA integrity number (RIN), have been generated for all samples (median medTIN=77.12). There were no statistically significant differences between medTINs from different comparisons (e.g. normal skin vs. BCC, ...) ($p > 0.05$; Mann-Whitney U test; Supplementary Figure 6)." Lines 365-370

R1-C3

Samples from Atwood and Sharpe may be overlapping. Was uniqueness of the sample origin verified?

Our response: We were aware of this possibility and thus have verified the uniqueness of samples prior to analyses. We considered only non-overlapping samples in all our analyses.

R1-C4

In general, most aggressive tumors are frequently bigger and with better tumor purity (TP) than benign lesions, this may explain "enrichment" of tumor markers associated with aggressiveness. A graph representing distribution of tumor purity (TP) would be useful. It is also possible to compare TP estimates based on RNA with TP from mutational data. TP is an important confounding factor which can impact sample clusterization. PCA may help to reveal and account for the TP component.

Our response: We really appreciate the insightful suggestion from Reviewer #1.

We agree that, in general, for most cancers, bigger tumors are easier to isolate and have higher tumor purity. In the case of basal cell carcinomas, some differences exist. First, small tumors can be easily demarcated, both clinically and with dermoscopy. Second, larger tumors are typically more inflammatory or may be ulcerated. Third, basal cell carcinomas rely heavily on stromal elements, and are in fact characterized by a fibromyxoid stroma, especially for high-risk BCC and BCC undergoing progression [Kaur et al., 2006]. Both an inflammatory response and a stromal response are characteristic [Omland, 2017]. Most gene expression-based algorithms for tumor purities consider tumor purity as pure tumoral component minus stromal abundance and immune infiltration [Yoshihara et al., 2013].

We have used the ESTIMATE algorithm [Yoshihara et al., 2013] to compute *in silico* tumor purities from expression data. As suggested by Reviewer #1, we have divided BCC into aggressive BCC (advanced tumors + high-risk histopathological subtypes) and non-aggressive BCC. Median tumor purities were 81.5% for aggressive BCC vs. 91.0% for non-aggressive counterparts, a statistically significance difference (Whitney-Mann U test, $p < 0.001$). We have added a paragraph in the methods about tumor purity and a Supplementary Figure 7.

We appreciate the value in using mutational data to compare tumor purity. Unfortunately, for most studies, mutational data were not readily available.

We applied the suggestion to account for tumor purity for clustering analyses, in addition to a batch effect. All heatmaps were updated to reflect these changes.

To better visualize effects from tumor purity, as suggested by Reviewer #1, we have performed Principal Component Analysis (PCA) on gene expression and tumor purity. We have generated PCA plots for all four comparisons (Supplementary Figure 8). Points on the PCA plots were sized according to tumor purity (higher tumor purity = larger size). We did not observe significant clustering of samples based primarily on tumor purity.

- Supplementary Figure 7

- Supplementary Figure 8

"Tumor purity was determined by ESTIMATE [Yoshihara et al., 2013] using gene expression data. We have divided BCCs into aggressive BCCs (advanced tumors + high-risk BCCs based on histopathological subtypes) and non-aggressive BCCs. Median tumor purities were 81.5% for aggressive BCCs vs. 91.0% for non-aggressive counterparts, a statistically significance difference ($p < 0.001$; Whitney-Mann U test; Supplementary Figure 7). Considering the biology of BCC, this is expected for several reasons. First, small tumors usually can be easily demarcated, both clinically and with dermoscopy. Second, advanced tumors are typically more inflammatory or may be ulcerated. Third, basal cell carcinomas rely heavily on stromal elements,

and are in fact characterized by a fibromyxoid stroma, especially for high-risk BCCs and BCCs undergoing progression [Kaur et al., 2006]. Both an inflammatory response and a stromal response are characteristic [Omland, 2017]. Most gene expression-based algorithms for tumor purities consider tumor purity as pure tumoral component minus (stromal abundance and immune infiltration) [Yoshihara et al., 2013].

To better visualize effects from tumor purity, we have performed Principal Component Analysis (PCA) on gene expression and tumor purity. We have generated PCA plots for all four comparisons according to the first two principal components, with points sized according to tumor purity (higher tumor purity = larger size) (Supplementary Figure 8). We did not observe significant clustering of samples based primarily on tumor purity." Lines 371-387

R1-C5

In figure 4c, high-risk BCC do not seem to form a separate cluster, low-risk BCC also represent one pure group, and other - includes both low and high risk BCC. There is no clear message about the differences at RNA level between these subgroups. How classification accuracy was calculated? Same comment is relevant for figure 5.

Our response: For classification accuracy, we have used V-measure [Rosenberg et al., 2007], which encompasses homogeneity of clusters and completeness of clusters. We have added this metric to all heatmap analyses.

We agree with Reviewer #1 that there is a mixed cluster with high-risk and low-risk tumors and not a true pure cluster. This is also the case, as pointed out, for advanced BCC and non-advanced BCC. We have expanded the discussion of clustering results to mention similarities in RNA expression of few genes that account at least in part for these mixed clusters.

"Overall group membership was 93.5%, with a V-Measure of 0.48 (homogeneity = 0.66, completeness = 0.38)." Lines 82-83

"Hierarchical clustering on high-risk vs. low-risk BCC samples based on expression levels of all 67 samples yielded 3 major patterns: one large cluster consisting only of low-risk BCCs, few early branching outgroups with high-risk BCCs, and a mixed cluster of high-risk and low-risk tumors (Figure 4A). The mixed cluster is comprised of samples with overall moderate upregulation of *SPHK1*, *MTHFD1*, *BMS1P20*, *PRMT6*, *ANGEL1*, *OLFML2B*, and *C1QTNF6*, without any being highly upregulated. Clustering metrics were lower (V-measure = 0.28 with homogeneity = 0.52 and completeness = 0.19), while accurate group membership was 93.2%." Lines 157-163

"Hierarchical cluster analysis was performed using the expression levels for the top 50 upregulated genes. Three major groups were observed: one large cluster consisting purely of

non-advanced BCCs, one early outgroup joined with a small cluster of three advanced BCCs, and a mixed cluster without strong sub-cluster associations between advanced and non-advanced tumors (Figure 5A). The mixed cluster is comprised of samples with moderate to moderate-high upregulation of *COL1A1*, *COL1A2*, *COL3A1*, *FNI*, and *LUM1*, without a strong *SFRP2* expression. Clustering metrics were lower (V-measure = 0.35 with homogeneity = 0.50 and completeness = 0.27), while accurate group membership was 87.3%." Lines 183-190

"Despite a limited number of differentially expressed genes, a group membership score of 100% was achieved when considering one large cluster only consisting of vismodegib-resistant tumors, and one looser association of an early outgroup of vismodegib-sensitive BCCs joined to a small cluster of vismodegib-sensitive BCCs. Clustering metrics were excellent, with a V-Measure of 0.84 (homogeneity = 1 and completeness = 0.72)." Lines 220-225

"V-measure, a clustering validity metric combining homogeneity of clusters and completeness of clusters, were determined for all heatmaps [Rosenberg et al., 2007]." Lines 408-410

R1-C6

One of the interesting findings is the upregulation of WNT pathway in vismodegib-resistant BCC. Would this result remain if vismodegib-resistant BCC contrasted with naïve to treatment BCC, majority of which are expected to respond to vismodegib? This would help to improve statistics.

Our response: We are grateful for this idea that adds robustness to our manuscript. By contrasting vismodegib-naïve untreated BCC (most of them being low-risk BCC) and vismodegib-treated, vismodegib-resistant BCC, we confirmed that the Wnt/ β -catenin pathway is also enriched among upregulated genes in vismodegib-resistant tumors ($p = 0.016$; Panther). The two novel Wnt/ β -catenin genes upregulated when comparing resistant to sensitive tumors, *DACT1* (Fold change=1.91; Q-value = 0.007) and *FSTL1* (Fold change=1.92; Q-value = 0.0002), were also upregulated in vismodegib-resistant BCC when using naïve tumors as a control set. We have added a paragraph in the results section and discussed the findings in the discussion section as well. We have added a panel to Figure 6 to present violin plots for both genes (*DACT1* and *FSTL1*) when comparing naïve to resistant tumors.

- Figure 6D (panel added)

"To provide additional robustness regarding the upregulation of Wnt/ β -catenin signaling in vismodegib-resistant BCCs, we compared vismodegib-resistant tumors to untreated, vismodegib-naïve BCCs instead of vismodegib-treated, vismodegib-sensitive BCCs. Low-risk BCCs represent the majority of vismodegib-naïve BCCs. Gene expression analyses and pathway enrichment analyses were performed as for other comparisons. Despite differences in biological

and clinical behavior between vismodegib-naïve and vismodegib-resistant BCCs, the upregulation of Wnt/ β -catenin signaling in vismodegib-resistant BCCs remained ($p = 0.016$; PANTHER). Overexpression of FSTL1 (Fold-change = 1.92; FDR-adjusted p -value = 0.0002) and DACT1 (Fold-change = 1.91; FDR-adjusted p -value = 0.007) in vismodegib-resistant BCCs persisted (Figure 6D; violin plots)." Lines 234-243

"The upregulation of Wnt/ β -catenin pathway and of two of its modulators, *FSTL1* and *DACT1*, in resistant tumors is likely an intrinsic characteristic of vismodegib resistance, as it remains elevated compared to untreated, vismodegib-naïve BCCs." Lines 311-313

R1-C7

Labeling of the figures does not correspond to references in the text.

Our response: We are truly sorry for our mistake and apologize for any inconvenience to the editors and reviewers. We have re-ordered all figure panels according to the references in the text.

Reviewer #2: We thank Reviewer #2 for bringing to our attention many essential points that needed clarifications and modifications.

This is a very interesting manuscript from an established group of investigators. The authors identified IL-17 pathways previously not shown in various clinical types of BCC. While there is a certain novelty of their research, there are several minor points that need to be addressed.

R2-C1

p. 5, the second paragraph, about lipid-related metabolic pathways and "this can be reflected in the tendency for BCC to ulcerate and/or easily bleed." This statement is purely hypothetical and should be substantiated by the appropriate references. I would recommend moving it to a discussion from the results.

Our response: We agree with Reviewer #2 that the statement in the current form was too speculative. We have rewritten these sentences with better substantiation with current literature. The tendency for BCC to ulcerate and bleed might be better explained by increased microvessel density with increased angiogenesis due to inflammatory cytokines and VEGF [Aoki et al., 2003]. However, alterations in lipid metabolism are more frequently associated with non-healing wound.

- Moved section from Results to Discussion

- It now reads:

"Among BCC tumors, lipid-related metabolic pathways that are essential to generate, maintain and regenerate a fully functional keratinized epidermis appear to be downregulated. For example, impaired lipid metabolism and downregulation of PPAR pathways are associated with impaired wound healing [Michalik et al., 2006]. These alterations might contribute to the non-healing phenotype of certain BCCs." Lines 271-275

R2-C2

All violin plots are lacking statistical analysis of the differences. Please provide the appropriate statistics on the charts.

Our response: We have added FDR-adjusted p-values/Q-values for all Violin plots directly on the charts.

R2-C3

p.6 "Validation in McGill BCC samples cohort." The authors identified 19 differentially-expressed genes that they try to validate. Traditionally, the validation involves ROC curves and AUC comparison between the previous set and new set of samples. It is tough to judge if those results are valid by a cluster analysis.

Our response: We believe this point needs clarification. The heatmap presented as a validation in our cohort of BCC samples is a simplified visual representation of all RT-qPCR experiments rather than a clustering analysis, using RNA-Seq data as the discovery stage. Colors refer to fold-enrichment for that particular gene-sample combination, after normalization. There are obvious technical differences between RNA-Seq and RT-qPCR that make it challenging to directly compare results.

Using a simple approach where >50% of upregulated genes should have >2-fold enrichment to be classified as defining the BCC phenotype/disease, this method yielded an AUC of 0.933, sensitivity of 0.87 and specificity of 1.00. This is now added to a Supplementary Figure 3 – panel A.

We have also added additional graphical representation of individual biomarkers as ROC curves, considering our cohort as a new set of samples. This is now Supplementary Figure 3 – panel B.

We also included a mention that a larger validation cohort is needed to further characterize these biomarkers.

- Supplementary Figure 3

"Overall, 13 out of 15 BCC samples show normalized enrichment above 2-fold for at least 50% of qRT-PCR-tested genes (heatmap summarizing results in Figure 3); this simple approach yielded a sensitivity of 0.87 and a specificity of 1.0 for diagnosing a BCC (Supplementary Figure

3A; AUC = 0.93). Receiver-Operator Curves for individual genes as potential biomarkers are presented in Supplementary Figure 3B." Lines 119-124

"A larger validation cohort is likely needed to further characterize these biomarkers." Lines 343-344

"Receiver-Operator Curves (ROC) curves were generated using R package pROC [Robin et al., 2011]." Lines 424-425

R2-C4

p.7 "High risk BCC with histopathological subtype requiring Mohs..."

"BCC tumors with superficial and nodular histopathological patterns are usually considered low-risk." This statement is incorrect. The nodular BCC are never considered to be a low-risk, and for that reason, none of the topical treatments are FDA approved for the treatment of nodular BCC. There are very strict criteria for Mohs selection of NMSC, which include not only histological type (by the way, basosquamous BCC is not one of them, but metatypical and keratotic is included), but also area of the body, the patient characteristics, and the positive margin on the recent excision/biopsy (Connolly et al AAD/ACMS/ASDSA/ASMS 2012 appropriate use criteria... JAAD, 2012). Thus, it is not entirely clear to me what criteria were used for high-risk BCC samples. Samples that were obtained from Mohs? Some histological features of BCC? In any case, the samples may be too heterogenous, and I am questioning the utility of such separation.

Our response: We thank Reviewer #2 for bringing to our attention these valid concerns. We have put greater emphasis on the wide clinical and histopathological presentations of BCC, on how features related to tumor characteristics, body area, and patient characteristics may portend a low risk or a high risk for local recurrence and aggressive tissue destruction, and on the importance of using the 2012 appropriate use criteria for Mohs. We have removed basosquamous and added keratotic from descriptions – none were represented among the sample set.

We have specified that, in this manuscript, we considered only the Mohs appropriate use criterion of "tumor characteristics – histopathological subtypes with aggressive features" [Ad Hoc Task force, Connolly et al., 2012] to define high-risk vs. low-risk BCC. This statement was included several times in the manuscript as a reminder.

We see value of this separation as a proxy for intrinsic biological behavior. Genomic studies on rare histological BCC subtypes are uncommon and we believe they can bring meaningful data to understand their underlying pathogenesis and guide their management in the future (alternative treatment options, combination with Mohs and medical therapies, etc.).

We have erased "Conservative" vs. "Mohs" on figures to remove any confusion.

"BCC tumors can present with a wide range of clinical and histopathological presentations, which may render them high risk for local recurrence and aggressive tissue destruction. Features such as body locations, patient characteristics (immunosuppression, genetic syndromes, radiotherapy, etc.) and tumor characteristics (histopathological subtypes with aggressive features, margin positivity, recurrent tumor, etc.) all have the potential to influence BCC risk level. The 2012 American Academy of Dermatology, American College of Mohs Surgery, American Society for Dermatologic Surgery Association, and the American Society for Mohs Surgery appropriate use criteria for Mohs micrographic surgery help identify clinical scenarios most likely to benefit from Mohs surgery, in the context of limited accessibility [Ad Hoc Task force, Connolly et al., 2012]. BCC with high risk features are usually managed by Mohs micrographic surgery, which was shown to significantly reduce the recurrence rate over more traditional medical and/or surgical approaches [16].

For subsequent analyses, in this manuscript, we consider only the Mohs appropriate use criterion of "tumor characteristics – histopathological subtypes with aggressive features" [Ad Hoc Task force, Connolly et al., 2012]. Thus, BCC tumors with superficial and nodular (especially on a low risk body area) patterns would be considered low-risk, when compared to high-risk BCC tumors with histological subtypes indicating a more aggressive biological behavior, such as morpheaform, infiltrating, metatypical, keratotic and micronodular.

In our analyses, we compared the transcriptome of six high-risk BCC samples with 38 low-risk BCC samples, based uniquely on the histopathological subtypes with aggressive features criteria from the 2012 appropriate use criteria for Mohs micrographic surgery (see Methods)." Lines 135-154

", 2) high-risk BCCs with histological subtypes having aggressive features (morpheaform, micronodular, metatypical, keratotic and infiltrative) vs. low-risk BCCs (nodular, superficial, others) 7," Lines 391-393

R2-C5

p.8 "Advances vs. non-advanced BCC" Please clarify the clinical and histological characteristics of your advanced BCC. What was the reason not to perform the surgery or radiation? Size of the tumors, age of the patients, location? All those features may contribute to the selection bias of your BCC in the advanced group

Our response: Detailed clinical information were limited in the 3 main studies from which RNA-Seq data originated. All advanced tumors satisfied the criteria that were used for FDA approval of vismodegib and sonidegib as we reported below:

"Advanced BCCs include locally advanced and metastatic tumors..... Although locally advanced BCCs do not have formal definite criteria, they represent tumors for which surgery and/or radiotherapy are usually not curative or not appropriate (e.g. resulting in mutilation) [9]."
Lines 174-178

We added a statement that all advanced BCCs satisfied these criteria:

"All advanced tumors satisfied the inclusion criteria given in the prior paragraph." Lines 179-180

The precise information why radiation or surgery was not performed on a per patient basis remains unknown. We have added a cautionary statement in the discussion about this limitation:

"Another limitation relates to missing key elements from the clinical information. For example, for advanced BCCs, it was not possible to know whether a selection bias might have been a reason why surgery and/or radiation were not performed, or whether patient and tumor factors such as the underlying location, immunosuppression status, and histopathological subtypes played a role in clinical decision making." Lines 284-288

R2-C6

p.9 how vismodegib-sensitive and resistant tumors were established? Ex vivo? Clinical response? please clarify

Our response: Clinical response was used to establish vismodegib resistance and sensitivity, as per RECIST guidelines (progressive disease or stable disease for resistance, any responder for sensitivity). We added this key information in the results section:

"Clinical response to vismodegib as defined by the Response Evaluation Criteria in Solid Tumors (RECIST) guidelines was the endpoint for vismodegib sensitivity (partial response or complete response) vs. resistance (progressive disease or stable disease) [Atwood et al, 2015][Sharpe et al., 2015]." Lines 214-216

R2-C7

p. 10 "for advanced BCC, anti-IL17 therapy may have activity..." the reference that was provided at the end of that sentence does not support this statement. Please, supply the correct reference.

Our response: We apologize that the end of the statement and the reference did not match the initial part. We have corrected the statement and added the relevant references. We also expanded the section on IL-17 as a therapeutic target for

advanced BCC with relevant studies in animals and in cell lines. We added that this remains to be proven in clinical trials.

"IL-17 and IL-22 cytokines have been directly implicated in BCC and SCC progression and tumor growth in mouse xenografts, as well as progression, cell migration and local invasion in BCC cancer cell lines [Nardinocchi et al., 2015]. Another report confirmed the upregulation of both cytokines in peritumoral skin of BCC, correlating with the severity of the inflammatory infiltrate [Pellegrini et al., 2017]. Both studies have suggested IL-17 as a potential therapeutic target. From our analysis, for advanced BCC, there might be a potential benefit using anti-IL-17 therapies for treating these tumors, with a favorable safety profile. This effect remains to be proven in clinical trials." Lines 257-264

R2-C8

p.11 discussion about natalizumab. I am not sure why there is a need to discuss JC virus reactivation in the manuscript about BCC

Our response: We have removed the section on Natalizumab given its lesser relevance to BCC as mentioned by Reviewer #2.

R2-C9

Figure 7A. The nodular BCC was shown on a histological picture, and the title underneath stated "conservative management (superficial BCC)". First of all, it is not a superficial BCC; second of all, the conservative management (whatever that means) is not appropriate for nodular BCC. The same goes for the next picture. The same BCC, the title underneath stated "simple BCC (superficial BCC)": again not superficial BCC on the picture but nodular and not sure what "simple" means. Please make some corrections.

Our response: We appreciate the astute observations brought by Reviewer #2. As discussed above, we have removed Conservative management and Mohs, and replaced them by low-risk BCC and high-risk BCC. We have replaced the picture of a nodular BCC by the picture of a superficial BCC. We agree that "Simple" BCC does not convey much meaning and is incorrect. We have changed "Simple" to non-advanced BCC to encompass BCC that do not satisfy the definition of advanced BCC. We have updated the Figure accordingly.

- Updated Figure 7A

Reviewer #3: We are glad Reviewer #3 enjoyed our manuscript and found our approach useful to identify actionable targets for BCC.

Lefrancois et al. perform an in-depth analysis of gene expression of basal cell carcinomas (BCC) to reveal novel pathways that may serve as targets for future treatment. The investigators studied a number of different BCC variant transcriptomes that include low-risk, high-risk, advanced, vismodegib sensitive and vismodegib resistant tumors. The team analyzed differences in gene expression utilizing RNAseq from publically available dataset. The combination of datasets yield robust findings for the different variants of BCC that highlight pathways important in the pathogenesis of BCC. The authors analyzed gene sets for relevant pathways using KEGG, PANTHER, Reactome and BioCarta. The genes identified revealed pathways that included IL-17, TLR, Akt/PI3K, cadherins, integrins, and the Wnt/beta-catenin pathways. A strength of the study is that the investigators were able to validated a subset of genes by qRT-PCR from primary BCC samples. The analysis is novel and innovative, and new targets for treatment of BCC may potentially be developed.

April 9, 2021

RE: Life Science Alliance Manuscript #LSA-2020-00651-TR

Dr. Philippe Lefrançois
McGill University Faculty of Medicine
Department of Medicine, Division of Dermatology
1650 Cedar Ave.
Montreal, Quebec H3G 1A3
Canada

Dear Dr. Lefrançois,

Thank you for submitting your revised manuscript entitled "Transcriptional landscape of basal cell carcinomas: novel signaling pathways and actionable targets". We apologize for this extended and unusual delay in getting back to you.

We were unable to secure the comments from the original set of referees. The revised manuscript was thus sent to experts on our editorial and advisory board, who assessed whether the authors had satisfactorily addressed all of the reviewers' concerns and whether it was now suitable for publication in Life Science Alliance (LSA). We are pleased to let you know that our experts have deemed "the authors have done a good job addressing the comments by the reviewers" and we would be happy to publish this in LSA pending minor revisions required to comply with LSA's formatting guidelines.

Along with the points listed below, please also attend to the following:

- please mark Dr. Lefrançois as Corresponding and Dr. Litvinov as 2ndary Corresponding Author. Currently, Dr. Lefrançois is listed both as Corresponding and 2ndary Corresponding Author
- please consult our manuscript preparation guidelines <https://www.life-science-alliance.org/manuscript-prep> and make sure your manuscript sections are in the correct order
- please add ORCID ID for the corresponding (and secondary corresponding) author--you should have received instructions on how to do so
- please add callouts for Figures 7A, B; S6 A-D; SA-D to your main manuscript text
- please add your main, supplementary figures, and table legends to the main manuscript text after the references section
- please add the Manuscript type when submitting the final version
- please add scale bars to the histology sections shown in Figure 7A
- the same histology section has been used in "low-risk BCC" and "non-advanced BCC" in Figure 7A. It is LSA's policy to not have duplicate images within the manuscript, could you please replace one of them with a separate image?

You will be guided to complete the submission of your revised manuscript and to fill in all necessary

information. Please get in touch in case you do not know or remember your login name.

A. FINAL FILES:

B. MANUSCRIPT ORGANIZATION AND FORMATTING:

Sincerely,

Shachi Bhatt, Ph.D.
Executive Editor
Life Science Alliance
<http://www.lsajournal.org>
Tweet @SciBhatt @LSAJournal

April 26, 2021

RE: Life Science Alliance Manuscript #LSA-2020-00651-TRR

Dr. Philippe Lefrançois
McGill University Faculty of Medicine
Department of Medicine, Division of Dermatology
1650 Cedar Ave.
Montreal, Quebec H3G 1A3
Canada

Dear Dr. Lefrançois,

Thank you for submitting your Research Article entitled "Transcriptional landscape of basal cell carcinomas: novel signaling pathways and actionable targets". It is a pleasure to let you know that your manuscript is now accepted for publication in Life Science Alliance. Congratulations on this interesting work.

DISTRIBUTION OF MATERIALS:

Again, congratulations on a very nice paper. I hope you found the review process to be constructive and are pleased with how the manuscript was handled editorially. We look forward to future exciting submissions from your lab.

Sincerely,

Shachi Bhatt, Ph.D.

Executive Editor

Life Science Alliance

<http://www.lsjournal.org>
